# The Interactivity between TGFβ and BMP Signaling in Organogenesis, Fibrosis, and Cancer

**DOI:** 10.3390/cells8101130

**Published:** 2019-09-23

**Authors:** Francesco Dituri, Carla Cossu, Serena Mancarella, Gianluigi Giannelli

**Affiliations:** National Institute of Gastroenterology “S. De Bellis”, Research Hospital, Castellana Grotte, 70013 Bari, Italy; carla.cossu@irccsdebellis.it (C.C.); serena.mancarella@irccsdebellis.it (S.M.); gianluigi.giannelli@irccsdebellis.it (G.G.)

**Keywords:** TGFβ, signaling, BMPs, SMAD pathway, Non-SMAD signaling, development, adult homeostasis, disease, fibrosis, cancer

## Abstract

The Transforming Growth Factor beta (TGFβ) and Bone Morphogenic Protein (BMP) pathways intersect at multiple signaling hubs and cooperatively or counteractively participate to bring about cellular processes which are critical not only for tissue morphogenesis and organogenesis during development, but also for adult tissue homeostasis. The proper functioning of the TGFβ/BMP pathway depends on its communication with other signaling pathways and any deregulation leads to developmental defects or diseases, including fibrosis and cancer. In this review we explore the cellular and physio-pathological contexts in which the synergism or antagonism between the TGFβ and BMP pathways are crucial determinants for the normal developmental processes, as well as the progression of fibrosis and malignancies.

## 1. TGFβ and BMP Pathways

Ample, detailed studies in the literature have described in great detail the multiple molecular mechanisms characterized to date that operate to transduce the signaling activated by secreted polypeptide growth factors belonging to the TGFβ superfamily. [1]. This large pool encompasses TGFβ, BMP, activin/inhibin, growth and differentiation factors (GDF), nodal, and the anti-Müllerian hormone, that are usually further assigned to families, based on structural and functional affinities among them [2]. The TGFβ family includes TGFβ1, TGFβ2 and TGFβ3, while more than ten BMPs have been defined in the relative family. The members of both families activate intracellular signals that can be grouped into two general signaling modes, namely the canonical and the non-canonical pathway. Both pathways initiate with the binding of TGFβ/BMP ligands to two types of receptors (type I and II) that then become active and drive the signal into the cytosol with the aid of various coreceptors. The canonical (also called SMAD-dependent) pathway uses a highly conserved simple linear cascade of SMADs (from 1 to 9) as signal transducers downstream of the receptors. On activation, the receptor complex phosphorylates the carboxy-terminus of receptor-regulated SMAD proteins (R-SMADs), including SMAD1, 5 and 8 for BMP signaling, and SMAD2 and 3 for TGFβ signaling. Activated R-SMADs interact with the common partner, SMAD4, and accumulate in the nucleus where the SMAD complex directly binds defined elements on the DNA and regulates the expression of numerous target genes. Inhibitor (I)-SMADs, namely SMAD6 and 7, have the task of regulating the R-SMADs cascade via various blockage mechanisms. Instead, non-canonical (or non-SMAD) signaling involves several components of other pathways, including Phosphatidylinositol 3-Kinase (PI3K)/Akt, Mitogen Activated Protein Kinases (MAPKs, namely Extracellular Receptor Kinase 1 and 2 –ERK1/2-, Jun N-terminal Kinase –JNK-, and p38), cell division control protein 42 homolog (Cdc42)/Rac and Rho-like GTPase [3,4,5,6,7,8]. Some of the principal features of TGFβ and BMPs signaling pathways are shown in Figure 1 and Table 1.

## 2. TGFβ and BMPs in Development

TGFβ and BMPs have long been of interest due to the functions they play in embryonic stem cells (ESCs) renewal, organ morphogenesis and development in many contexts, including the heart and skeletal system [23]. In mouse embryogenesis studies, it was hypothesized that one mechanism by which the TGFβ pathway may regulate the expression of genes promoting ESCs homeostasis or differentiation probably depends on SMAD2/3 preferential assembly with SMAD4, or with Tripartite Motif protein 33 (TRIM33) and other factors, to form differentially functioning transcriptional complexes [24]. While TGFβ and FGF are required for the maintenance of the human ESCs pluripotency state [25], BMP4, in the absence of TGFβ and Fibroblast Growth Factor (FGF), prompts the differentiation of these cells towards the formation of primitive trophoblasts ectoderm [26,27]. TGFβ counteracts the action of BMP2,4,6,7,9 on differentiation of mesenchymal stem cells (MSCs) into osteocytes and adipocytes, although TGFβ1 and TGFβ3 can cooperate with BMP2 to induce the differentiation of MSCs into chondrocytes [28,29,30,31]. TGFβ and BMP4 coordinately initiate and modulate a spatiotemporal pattern of activation of canonical SMADs (SMAD1, 2, 4, 5, 8) to control bladder development in mouse [32]. The degree of structural affinity shared by members of the TGFβ superfamily results in the binding of some BMPs to TGFβ receptors, and the consequent activation of SMAD2/3. Holtzhausen et al. observed that BMP activated SMAD2 is involved in the development of the dorsoventral axis in zebrafish embryos, whereas BMP activated SMAD3 contributes to promoting the invasion of transformed cells [33]. The following sections describe the role of TGFβ and BMPs in the development of bone and heart, taken as paradigmatic examples of how these cytokines interact in developmental processes.

### 2.1. Bone Morphogenesis

Bone morphogenesis requires a timely and spatially tuned intervention of both the TGFβ and BMPs pathways, in combination with other pathways such as MAPK, Wnt, Hedgehog (Hh), Notch, Akt/mTOR, and miRNA in a tremendously complex interplay [34,35]. TGFβ is released and activated by osteoclasts in the site of bone resorption and promotes the recruitment of MSCs [36], that then differentiate into osteoblasts as a result of contact with bone matrix. TGFβ enhances the production of FGF2 in osteoblasts [37], that in turn, stimulate proliferation of endothelial cells, which are recruited in the process of bone remodeling [38,39]. While BMP favor post-natal bone formation by promoting the differentiation of chondrocytes and osteoblasts, instead, TGFβ/activin signaling functions in a chondroinductive manner to inhibit the mineralization of the extracellular matrix. It follows that a translational potential in the treatment of osteochondral defects may arise following the use of TGFβ and specific BMPs to promote hyaline articular cartilage and subchondral bone regeneration, respectively [40,41]. However, BMP signaling can also indirectly counteract the bone matrix generation. More specifically, the BMP-driven activation of SMAD1/5-dependent pathway in osteoclasts limits the release of coupling factors, such as Wnt-1, Gja1 and Sphk1, that stimulate osteoblast mineralizing activity. Conversely, TGFβ counteracts this effect, through promoting Wnt-1 release [42]. TGFβ1 can enhance osteogenic functions mediated by BMP2. One mechanism involves the TGFβ induced increase of the expression of BMP2 receptor bone morphogenetic protein receptor type IB (BMPR-IB) in osteoblasts [43].

It has been established that TGFβ and BMP2 are required for normal mandibular development in mice. Interestingly, the activity of the transferrin receptor in neural crest cells was suggested to be necessary to favor normal surface expression of the TGFβ and BMP receptors by switching the internalization phase of their recycling paths from a faster caveolin- to a slower clathrin-dependent endocytosis [44]. TGFβRII and BMPR1A transduce organogenic signals of TGFβ and BMPs during embryonic formation of the teeth and palate by using SMAD4 and p38 activities in a functionally redundant manner [45]. The development of the posterior palatal shelf of mice appears to be finely regulated by a mechanism based on the inhibition of the canonical BMP cascade by that of TGFβ. Using animals with conditionally depleted Tgfbr2 in the palatal mesenchyme, it was determined that the limited SMAD4 is preferentially sequestered in the complex with phospho (p)-SMAD2/3 rather than p-SMAD1/5/8 [46]. The TGFβ/Wnt pathway interplay induces chondrocyte and osteoblast differentiation. For this purpose, TGFβ and Wnt signaling enhance each other’s receptors/ligands expression, and the β-catenin-independent Wnt pathway deviates that of TGFβ towards a SMAD1/5/8 dependent route [47,48,49,50]. In mice with osteoblast-specific Tgfbr2 CKO, an increased trabecular bone mass and decreased cortical bone mass were observed, a consequence of the missed phosphorylation of Parathyroid hormone (PTH) receptor (PTH1R) by TGFβR2. This signaling event is necessary to promote the endocytosis of TGFβR2 and PTH1R, that in turn prevents PTH from inducing BMP2 expression in osteoblasts, that mediate the anabolic function of PTH signaling in bone [51,52]. Even though TGFβ promotes differentiation of mesenchymal stem cells (MSC) into vascular smooth muscle cells (VSMC) [53,54], it prevents subsequent phosphate-stimulated VSMC differentiation into osteogenic-like cells. Mechanistically, phosphate induces the expression of BMP2, which then leads to SMAD1/5/8 and Wnt/β-catenin activation, to finally carry out osteogenic differentiation [55,56], whereas TGFβ offsets this effect through blocking SMAD1/5/8 and β-catenin nuclear translocation [57]. A simplified scheme of interactions between TGFβ and BMPs in bone development is shown in Figure 2a.

### 2.2. Heart Development

An intricate crosstalk involving Notch, BMP and TGFβ signaling drives the development of cardiac septa and valves. The successful development of these structures requires the function of receptors ALK2 (ACVR1), ALK3 (BMPR1A) and their ligands, which include TGFβs, activin and multiple BMPs. [58]. TGFβ and BMP pathway interact at the level of SMADs phosphorylation in endocardial endothelial cells, given that ALK2 loss in these cells leads to a reduced activation of TGFβ-related SMADs, as well as BMP-related SMADs [59]. TGFβ3 and BMP2 act in concert to induce the EMT in atrioventricular (AV) endocardial cells, that in this way gain the ability to invade the cardiac jelly to form the cardiac cushions as a prior step in valve formation [60,61]. It was further demonstrated that myocardium-derived BMP2 induces the expression of Notch1 and TGFβ2 in endocardial endothelial cells, and that the integration among BMP2, Notch1, and TGFβ2 signaling is required for these cells to accomplish EMT and invade the prospective valve territories. [62,63,64]. ALK3 expression in cardiac myocytes is required for the formation of trabeculae, compact myocardium, interventricular septum, and endocardial cushion. TGFβ2 produced by these cells in response to BMP2 binding to ALK3 is one of the main mediators of cushion morphogenesis [65]. An activated endocardial Notch pathway is required for TGFβ2 and TGFβRII expression in the AV channel myocardium [66]. Notch, TGFβ and BMP2 synergistically up-regulate the expression of the transcriptional repressor Snail in endocardial endothelial cells, an event claimed as critical for the EMT that guides the motion of these cells into cardiac jelly to form heart cushions in the AVC [64,67]. Protein interaction between GATA4 and SMAD4 are essential for correct development of atrioventricular septa and valves as demonstrated by pathological conditions related to some missense mutations of GATA4 (G303E and G296S), or GATA4 and SMAD4 haploinsufficiency [68]. A simplified scheme of involvement of TGFβ and BMP in heart valve formation is shown in Figure 2b.

## 3. Role of TGFβ-BMP Pathway Crosstalk in Fibrosis

A large body of evidence has suggested that TGFβ is among the most prominent inducers of fibrotic processes. The main knowledge on the crosstalk between TGFβ and BMPs during fibrosis comes from studies of organs most often affected by this disease, including kidneys, liver, lungs and heart. Fibrosis occurs as a result of many chronic inflammatory diseases, and is defined by the accumulation of excess extracellular matrix components (ECM) like collagen and fibronectin. 

If the tissue injury is severe or repetitive, or if the wound-healing response itself becomes dysregulated, the fibrotic process can lead to organ malfunction and death [69]. The main causes of fibrosis in industrialized countries include genetic disorders, persistent infections, recurrent exposure to toxins, irritants or smoke, chronic autoimmune inflammation and myocardial infarction. The initiating event common to all fibrotic diseases is the activation of ECM-producing myofibroblasts [69,70]. TGFβ production correlates with the progression of liver, lung, kidney and cardiac fibrosis and its inhibition has been shown to reduce the development of fibrosis in many experimental models. The TGFβ dependent induction of fibrosis relies on activation of both canonical and non-canonical signaling modes [71]. Due to structural similarity and shared signal transduction modality [72], BMPs are likely to interactively operate with TGFβ not only in development but also in the context of fibrogenesis. The TGFβ and BMP pathways regulate processes that predispose to fibrosis, such as the EMT, by widely interacting to either reciprocally enforce or hinder each other. The balance between the activation of TGFβ-dependent SMADs (SMAD2/3), or those that are BMP-dependent (SMAD1/5/8), is modulated by a competition for the assembly with SMAD4, along with other regulatory mechanisms, including the inhibitory action SMAD7 that downregulates TGFβ1R and SMAD2 [73,74]. Despite this antagonism, SMAD2, in the same way as SMAD1/5/8 and SMAD7, was found to play a protective role against hepatic and renal fibrosis [75,76], even though TGFβ and BMPs can also synergistically activate different intracellular non-SMAD mediators (such as HIPK2 and TAK1) to activate p38 and JNK and thereby cause apoptosis, or alternatively Limk1 and Limk2 to inhibit cofilin, an event that results in F-actin polymerization, that in turn contributes to EMT [77]. It is noteworthy that BMP ligands are also able to bind and activate TGFβ receptors. In this way, BMP2, BMP11 and BMP16 can induce SMAD2 and SMAD3 phosphorylation [78].

Considering these findings, the activity for BMPs results prevalently anti-fibrotic, often in antagonism with that of TGFβ, while fewer studies support an opposite trend. Indeed, according to Katsuno et al., BMP signaling can strengthen that of TGFβ, through activating the protein arginine methyltransferase 1 (PRMT1), which in turn methylates SMAD6/7, thereby enabling SMAD1/3/5 activation, to finally promote EMT during fibrosis as well as maintain a stem-like phenotype of cancer cells in malignancies [79].

### 3.1. TGFβ-BMP pathway in Liver Fibrosis

Hepatic fibrosis is a virtually inexorable degenerative process which develops as a consequence of underlying chronic liver injury triggered by various etiologies. It initiates with a cascade of biochemical and biophysical changes in the liver microenvironment, causing necrosis and apoptosis of hepatocytes and liver sinusoidal endothelial cells (LSECs), through the release of inflammatory mediators and profibrotic cytokines [80].

The main feature of fibrosis is the increased deposition of extracellular matrix (ECM) molecules by activated hepatic stellate cells (HSCs), as an attempt to limit the consequences of chronic parenchymal injury. HSCs transdifferentiate from quiescent cells to myofibroblast-like cells, that express α-smooth muscle actin (αSMA) and secrete extracellular matrix materials (collagens, fibronectin, laminin and proteoglycans) [81]. In an iterative carbon tetrachloride model of rat liver fibrosis it was demonstrated that alterations in the LSECs phenotype (lack of fenestration and capillarization), HSCs activation, and excessive accumulation of ECM proteins results in a dramatic change in the mechanical microenvironment, particularly in the matrix stiffness [82] and that this phenomenon contributes to the progression of liver fibrosis [83]. Activated HSCs, but also Kupffer cells, macrophages and platelets produce TGFβ1 which then activates more HSCs, that then produce and release ECM molecules, tissue metalloproteinases (MMPs, such as MMP2,9,13) and MMPs inhibitors (such as TIMP1) [84,85].

TGFβ plays a key role in fibrotic transformation of liver. It confers HSCs a proliferative and migratory capacity and promotes their differentiation into myofibroblasts, that hence produce extracellular matrix molecules. [86] Multiple positive or negative modulators of pro-fibrotic TGFβ signaling have been characterized over time. The miRNA miR-942 was shown to activate the pro-fibrotic program in HSCs, through downregulating the TGFβ signaling inhibiting pseudoreceptor BAMBI. Interestingly, miR-942 expression is induced by TGFβ itself and LPS via SMAD2/3 and NF-κB/p50 mediated pathways, respectively [87]. Chen and colleagues found that exogenously administering recombinant BMP7 can reduce the histopathological features of fibrotic lesions in a model of hepato-schistosomiasis-induced fibrosis. This effect was associated with diminished expression of TGFβ1, p-SMAD2, and the marker or differentiation of pro-fibrotic myofibroblasts, αSMA [88]. In another animal model of liver fibrosis induced by administering porcine serum it was shown that recombinant human (rh) BMP7 significantly reduces collagen deposition. Consistently with in vivo results, rhBMP7 significantly reduced the expression of collagen type-I and -III by primary HSCs via blocking the nuclear translocation of SMAD2/3 [89]. Moreover, a higher BMP2/TGFβ expression ratio was found in normal human and mouse hepatic tissues, whereas the amount of TGFβ resulted in increased fibrotic livers of animals subjected to surgery for bile duct ligation or carbon tetrachloride administration. Consistently, the authors highlighted a reciprocal negative control mechanism wherein TGFβ diminishes the expression and release of BMP2 by HSCs. BMP2 can, in turn, down-regulate TGFβ effects in the same cells via attenuating the expression of the ligand itself and its own receptors, thus resulting in a reduction in TGFβ1 dependent SMAD3 phosphorylation, as well as the expression of αSMA, fibronectin, and more generally, EMT markers [90].

Recently, analysis was made as to whether adenovirus-mediated overexpression of BMP7 antagonizes the effect of TGFβ in vitro and in vivo. Indeed, induction of BMP7 decreased the expression of collagen and αSMA mRNA via SMAD 1/5/8 phosphorylation in both primary cultured rat stellate cells and in human stellate cell line LX-2, and suppressed thioacetamide (TAA)-induced liver fibrosis in rats [91]. BMP7 antagonizes TGFβ signaling by competition of phospho-SMAD 1/5/8 and SMAD3 for specific DNA binding and/or recruitment of co-factors [32]. The reciprocal modulation of TGFβ/BMP interplay also has a strong impact on fibrotic progression associated with other pathological conditions, such as adipose tissue inflammation and non-alcoholic fatty liver disease (NAFLD). Gain/loss-of-function studies of kielin/chordin-like protein (KCP) determined that TGFβ and activins enhance, while BMP probably counteract, the pathological features of NAFLD in high fat diet-treated mice. Indeed, KCP overexpression in liver caused a shift in the pattern of SMAD activation from p-SMAD3 toward p-SMAD1 [92]. See Figure 3b for mechanistic aspects of fibrosis development in liver.

### 3.2. TGFβ-BMP Pathway in Renal Fibrosis

Renal fibrosis is the result of an imbalance between excessive deposition and decreased breakdown of the ECM in the interstitial compartment, that leads to scar formation. The deposition of ECM in a normal and fibrotic kidney follows the transition of interstitial fibroblasts to αSMA positive myofibroblasts. Tubular epithelial cells are the elements primarily exposed to damage during kidney injury [93].

The accumulation of ECM proteins is mainly related to the activity of TGFβ, whose overexpression in the kidneys of experimental animals is closely linked to renal fibrosis [94].

In studies using in-vitro and in-vivo models (renal fibroblasts and mice with renal fibrosis induced by ureteral ligation, respectively) TGFβ was found to promote a pro-fibrotic phenotype primarily via SMAD3 activation, while a more indirect mechanism is also involved. Namely, by means of signals evoked by intracellular reactive oxygen species (ROS), this cytokine induces phospho-activation of EGFR and p53 that, in turn, up-regulate disease-causative plasminogen activator inhibitor-1 (PAI-1) and connective tissue growth factor (CTGF). Interestingly, PAI-1 and CTGF induction requires both EGF-activated Erk2 and a synergic interaction between EGFR and TGFβR signaling to phospho-activate SMAD3 [95].

The BMPs are known to antagonize the pro-fibrotic aptitude of TGFβ in chronic nephropathies. Activation of the BMP pathway protects against kidney fibrosis and chronic kidney disease. In particular, BMP7 has been reported to counteract progressive chronic renal injury via blocking TGFβ1-promoted EMT [96]. More specifically, TGFβ1-activated SMAD3 mediates renal inflammation and fibrosis via up-regulating miR-192 and miR-21, and down-regulating miR-29 and miR-200. These effects can be attenuated by the renoprotective BMP7, that, by means of SMAD1/5/8, upregulates miR-200 and miR-29, that reverses the pro-fibrotic action of TGFβ [75]. A mechanism was defined in human renal proximal tubular epithelial cells (HK-2 cell line), wherein BMP7 offsets the TGFβ-mediated loss of the SMAD transcriptional corepressor –SnoN, thus preventing SMAD3 (but not SMAD2) from binding to the promoters of its downstream-regulated genes. [97]. Other researchers have come to similar conclusions, although the mechanisms implicated differ in some aspects. In cultured mesangial cells BMP7 was shown to antagonize the induction of fibrosis by TGFβ, via a mechanism requiring SMAD5 and its downstream SMAD6, that impair the nuclear accumulation of SMAD3, and subsequent up-regulation of CAGA-lux and secretion of plasminogen activator inhibitor type 1 –PAI-1. Therefore, the diminished expression of PAI-1 consequent to BMP7 stimulus unleashes MMP2 activity, that then degrades extracellular matrix molecules, including collagen type IV and fibronectin, soluble collagen type IV, and thrombospondin [98,99]. 

BMP7 can also bind to activin–like kinase 3 receptor (Alk3) on tubular epithelial cells, and exert anti–inflammatory and anti–apoptotic functions. The assumption that Alk3 can favor renal health during kidney injury prompted Sugimoto et al. (2012) to develop a small synthetic molecule BMP7 mimetic, the peptide agonist THR–123. Notably, this compound was found to be able to bind to Alk3 and activate the SMAD pathway, ultimately delivering the same anti-fibrotic activity as recombinant BMP7 in a mouse model of kidney disease [100].

Altogether, these data have prompted consideration of the use of BMPs to ameliorate chronic fibrotic diseases, such as nephropathies, especially by virtue of their ability to counterbalance the pro-fibrotic action of TGFβ [101]. Due to limited clinical use of BMPs related to costly production or hazardous doses required to achieve efficacy, some synthetic BMP agonists or inducers are being tested in preclinical models. Following small-molecule high-throughput screening for synthetic compounds, the benzoxazole compound BMP agonist, sb4, has shown in vitro efficacy in stimulating BMP signaling and is thus a promising prospect for the treatment of chronic kidney disease [102]. Peritoneal dialysis offers an option in the treatment of this illness but this practice causes mesothelial cells to lose MBP7 expression and undergo EMT conversion which in turn leads to inflammation, fibrosis, angiogenesis and ultimately peritoneal membrane damage. BMP7 administration has proven to reduce this detrimental outcome in mice models of peritoneal dialysis [101]. See Figure 3a for mechanistic aspects of fibrosis development in kidneys.

### 3.3. TGFβ-BMP Pathway in Pulmonary Fibrosis

Pulmonary fibrosis includes a heterogeneous group of lung disorders characterized by the progressive and irreversible destruction of lung architecture characterized by the persistence of a fibrotic scar as the result of accumulation of extracellular matrix components at the site of tissue injury. The normal wound healing process following lung damage includes (1) a coagulation phase; (2) an inflammatory cell migration phase; (3) a fibroblast proliferation and activation phase; (4) a final remodeling phase where normal tissue architecture is restored. Fibrosis can develop if any of these stages is dysregulated. [103]. TGFβ is considered the major pro-fibrotic cytokine necessary not only for lung organogenesis and homeostasis but also for achieving trans-differentiation of lung fibroblasts to myofibroblasts during pulmonary fibrosis [104]. In lungs, BMPs play a protective role against the fibrosis-promoting activity of TGFβ. In bleomycin-induced lung fibrosis, a dynamic activation of TGFβ signaling and a repression of BMP signaling activity was observed [105]. BMP4 and BMP7 can attenuate the production of extracellular matrix proteins (tenascin C, fibronectin and collagen type I and IV) and metalloproteinases MMP2, respectively, in response to TGFβ by normal lung fibroblasts. Both BMP4 and BMP7 also reduce the TGFβ-induced production of MMP13 by the same cells. BMP7, but not BMP4, can also partially revert the TGFβ induced myofibroblast-like transformation of lung fibroblasts [106]. The disturbance of this fine TGFβ/BMPs balance may result in fibrosis. In a mouse model of asbestos-induced pulmonary fibrosis, an increase of gremlin was observed, which was demonstrated to cause BMPs downregulation and subsequently enhanced profibrotic TGFβ-dependent SMAD2 activation. Amelioration of the disease (evidenced by reduced hydroxyproline contents) was observed following the rescue of BMP signaling through BMP7 administration or using inhibitors of TGFβR [107].

The anti-fibrotic agent pirfenidone is used to treat idiopathic pulmonary fibrosis and may be useful as adjuvant therapy for mesothelioma, due to its ability to stimulate the BMP pathway while impairing AKT/mTOR signaling, non-canonical Akt and Erk dependent TGFβ signaling, and expression of the BMP antagonist gremlin-1 (GREM1) [108]. Notably, GREM1 expression is induced by the pro-fibrotic cytokine IL-6, that is elevated in multiple fibrotic conditions [109]. BMP7 was proven to decrease silica-induced pulmonary fibrosis through modulation of the balance between suppression of TGFβ/SMAD and activation of the BMP7/SMAD signaling pathway. More specifically, BMP7 attenuates the decrease of p-SMAD1/5 and the increase of p-SMAD2/3, as well as inhibiting RLE-6TN cells migration and expression of fibrosis markers (such as collagen I and III) [110]. See Figure 3b for mechanistic aspects of fibrosis development in lungs.

### 3.4. TGFβ-BMP Pathway in Myocardial Fibrosis

Myocardial fibrosis is a multiphase reparative response to cardiac ischaemic insults, systemic diseases, drugs, or any other harmful stimulus in the circulatory system. It is characterized by dysregulated collagen turnover and alteration of the architecture of the myocardium. The damaged tissue is replaced with a fibrotic scar produced by fibroblasts phenotypically transformed into myofibroblasts [111].

The most prominent characteristic of myofibroblasts is their migratory and contracting phenotype, which results from the expression of contractile proteins such as α-smooth muscle actin (αSMA). Although activated myofibroblasts are the main effector cells in remodeling of the fibrotic heart, several cell types (endothelial cells, cardiomyocytes, monocytes/macrophages and mesenchymal cells) are also implicated, and can act directly by producing matrix proteins, or indirectly by secreting fibrogenic mediators [112].

BMP2 has proven capable of neutralizing the TGFβ1 action causing cardiac fibrosis. More specifically, in cardiomyocytes BMP2 promotes the formation of the SMAD6/Smurf1 complex, that reverses the pro-fibrotic effects of TGFβ1 by promoting the degradation of its receptor and impairing the TGFβ-dependent activation of SMAD3 and Rho-associated kinase (ROCK). Of note, the latter is able in turn to suppress BMP2 expression through activating PKCδ [113].

Recently, a study suggested that BMP7 may counteract myocardial fibrosis through limiting the TGFβ1/SMAD3 dependent down-regulation of E-cadherin and up-regulation of collagen I and αSMA in myocardial fibroblasts. It was determined that this effect requires the activation of SMAD1/5 [114]. BMP7 was proven to limit myocardial fibrosis via attenuating TGFβ1 signaling also in the rat acute myocardial infarction model (AMI). Mainly, exogenous administration of recombinant BMP7 decreased TGFβ1 signaling in different areas of the left ventricle, as well as the myocardial fibrosis level and infarct size. However, molecular aspects involved in this mechanism are still poorly understood [115]. See Figure 3b for mechanistic aspects of development cardiac fibrosis.

## 4. Crosstalk between BMP and TGFβ Pathway in Cancer

The relevance of TGFβ signaling as a contributor in the progression of many cancers is extensively documented, but at the same time widely debated, mainly due to its context- and micro-environment dependence, and frequently conflicting downstream effects [1,116,117,118,119,120]. Indeed, both tumor promotion and suppressive functions have been described for this cytokine. TGFβ was suggested to induce cryostasis in the early stages of hepatocellular carcinoma (HCC), whereas it might work to support pro-invasive processes such as the EMT, invasion and fibrogenesis in the late phase [121]. Regardless of the HCC stage, the anti-tumor activity of TGFβ is likely related to its ability to induce the expression proteins that block the growth of cancerous cells, such as LATS1 and YAP1 [122]. Moreover, by clustering subtypes of HCCs according to mutational inactivation or hyper-activation of genes of TGFβ pathway components, Chen and colleagues found that survival was worse in patients with inactivation than those with hyper-activating gene alterations of the TGFβ pathway. Interestingly, the first group showed a loss of function of tumor suppressor genes, including those for oxidative stress response and DNA damage repair, whereas various proto-oncogenes resulted more activated in the second [123]. Nevertheless, in several cancer types, albeit predominantly gastrointestinal cancers and HCC, metastasis occurrence resulted increased and survival reduced in patients bearing TGFβ pathway activating mutations, as compared to subjects with the non-mutated TGFβ pathway [124].

The role of BMPs in cancer is no less controversial than that of TGFβ. Aberrant expression of BMPs or mutations of elements belonging to their related pathways have been described in diverse malignancies, including HCC, lung and gastric cancer [125,126,127,128]. BMPs may either act as tumor supporters, via promoting motility and invasiveness of some cancer cells, including prostate, colon and melanoma, or, on the contrary, limit the pro-metastatic aptitude of other cell lines, such as breast cancer cells. BMP9 was found to stimulate the proliferation of ovarian cancer cells, whereas it induces apoptosis in prostate cancer cells [129,130,131,132,133,134,135]. In an immunocompetent syngeneic orthotopic spontaneous mouse model of breast cancer (E0771) BMP9, but not BMP10, was shown to exert a protective action against tumor growth and lungs dissemination, probably via inducing quiescence in cancer cells [136]. However, the induction of cancer cells dormancy may turn into a double-edged weapon as it only represents a temporarily suspended malignant progression that may desensitize these cells to chemotherapy [137]. Some BMPs, such as BMP7 and BMP4, can induce dormancy in prostate and mammary carcinoma cells, respectively. The notion that BMP4 enforces tumor dormancy is supported by its ability to block in-vitro tumor sphere formation, to diminish the expression of Nanog, Sox2, and Taz, and increase that of GATA3 [138]. BMP7-induced dormancy requires the (BMP receptor 2), and is related to activated p38 mitogen-activated protein kinase and increased expression of the cyclin dependent kinase inhibitor, p21, along with the metastasis suppressor gene, *NDRG1* (N-myc downstream-regulated gene 1) [139].

In a context-dependent manner, TGFβ1/2 and BMP4/7 related cascades can cooperatively induce tumor cell dormancy via activating both canonical SMAD-regulated and non-canonical p38 MAPK-regulated gene expression patterns, but also through up-regulating cyclin dependent kinases inhibitors p21 and p27, and blocking ERK. Interestingly, TGFβ1, but not TGFβ2, can reawaken cells from dormancy by exploiting SMAD and PI3K/AKT dependent axes, that inhibit p16 while enhancing the expression of pro-proliferative factors, including Kiel-67 (KI67), Hairy and Snhancer of Split-1 (HES1), and MYC [140]. The ability of some BMPs to contrast the pro-metastatic aptitude of TGFβ was documented both in in-vitro, and in animal models of prostate and breast cancer. Naber et al. found that BMP7, but not its closely related homolog BMP6 can offset the pro-invasive activity of TGFβ in invasion tests using spheroids made from the metastatic breast cancer cell line MCF10CA1a, by a mechanism that inhibits the expression of the integrin αvβ3 [141]. Interestingly, BMP7 downregulation in human primary breast cancers is clinically correlated with a tendency to develop bone metastases. Systemic administration of BMP7 in breast cancer mouse models has proven to significantly inhibit bone metastases originating from human prostate and breast cancer, by reducing TGFβ-dependent SMAD3/4 activation, activating nuclear signaling of SMAD1/4/5, and restoring E-cadherin levels. In addition, BMP7 turned TGFβ into an accomplice in enhancing BMP signaling [142,143]. The BMP and TGFβ signaling can be differentially modulated by TGFβ coreceptors, or act in a synergic or mutually inhibitory manner in different malignancies. In breast cancer cells the membrane type III TGFβ coreceptor (TβRIII) is highly expressed, and can inhibit TGFβ or BMP signals via at least two different modes: (1) by shedding to form a soluble sTβRIII that can sequester the BMP ligand and prevent subsequent membrane binding and SMAD1/5/8 activation [144]; (2) by complexing with TβRI or TβRII and competing with TβRI/TβRII complex formation [145]. TβRIII, in turn, was found to be down regulated by TGFβ and BMP4 in ovarian, breast and pancreatic carcinoma cells. This event was shown to be necessary for these ligands to induce the EMT [146,147].

The reciprocal antagonism between TGFβ and BMP signaling occurs at multiple hubs. According to Ning et al., TGFβ binding to TβRI/II can inhibit the activation of the BMP receptor via enhancing the expression of CTGF and gremlin, and the activation of SMAD1/5/8 by upregulating Tmeff1 and miR-155. Conversely, BMPs binding to BMPRI/I can block the activation of TGFβ receptors via upregulating CTGF itself and SMAD6. Moreover, one pathway can overcome the other when competing for the use of SMAD4 [148]. The type I membrane glycoprotein endoglin was found to act as a modulator of the relative impact of the TGFβ and BMP axes on malignancy in Ewing sarcoma and melanoma cells. Through blocking the activation of TβRI by TGFβ and enhancing that of BMPRII by BMP2/4, endoglin can promote a higher tumor cell plasticity, also thanks to its ability to more indirectly activate other signaling mediators, including integrin β3, osteopontin, PI3K and focal adhesion kinase (FAK) [149]. Romagnoli et al. found that BMP5, which binds to type I and II BMP receptors, prevents the EMT through inhibiting the expression of the transcription factors Snail, that promote the repression of E-cadherin. However, TGFβ can override the block of EMT by BMP-5 in breast cancer cells, in a SMAD-independent way. Specifically, these authors identified a signaling process involving cRAF, AP1 and Blimp-1, which can restore the expression of Snail through suppressing that of BMP5 [150]. BMP7 can reverse TGFβ promoted migration and EMT phenotype in cholangiocarcinoma cells [151]. BMP and TGFβ signaling can antagonistically regulate the transcription of oncogenes, such as Cripto-1 (CR-1) gene. CR-1 is a membrane protein that was found to be abnormally accumulated in several different types of human carcinomas. It promotes cell proliferation and migration via activating PI3K/Akt and Ras/Raf/MAPK axes. It was found that TGFβ up-regulates, whereas BMP4 down-regulates the expression of CR-1 in embryonal carcinoma cells and colon cancer cells. [152,153]. Moreover, BMP4 function as tumor suppressor in in-vitro and orthotopic models of glioblastoma multiforme, where it limits tumor cell migration and invasion via SMAD1/5/8 mediated enhancement of E-cadherin and claudin expression [154]. The relative importance of the BMP and TGFβ pathways in promoting or suppressing the formation of tumors in in-vivo experimental settings may also depend on the type of cancer and/or signaling mode used. Taguchi and colleagues reported that in clear cell renal carcinoma (ccRCC) the signaling of TGFβ, but not that of BMP, works in a tumor-suppressive mode, by inducing apoptosis. Consistently, the transcriptional corepressor c-Ski, which is highly expressed in ccRCC, promotes tumor growth by counteracting the TGFβ-dependent SMAD arm (that requires SMAD2/3), but probably not that related to BMP (that requires SMAD1/5/8) [155]. However, in another study, BMP9 was shown to inhibit the proliferation of HCC cell lines via inducing, in a SMAD-independent mode, the expression of cell cycle inhibitor p21. Consistently, the expression of liver cancer stem cell markers, including CD44, CD90, AFP, GPC3 and ANPEP resulted suppressed by p21 [156]. A partial interactive network involving TGFβ and BMP actions in cancer progression is shown in Figure 4.

## 5. Conclusions

Evidence accumulating over recent decades has demonstrated the importance of the TGFβ and BMP signaling cascades in the completion of physiological processes such as organogenesis, but also in pathological conditions, mainly fibrosis-related disorders and various neoplasms. The complex interplay between the TGFβ and BMP pathways, along with their pleiotropic nature, can result in reciprocally enhanced or else antagonized effects in a context-dependent manner. It follows that the design of more promising anti-TGFβ/BMP pharmacological strategies to treat fibrotic and cancerous diseases is a challenging task that will undoubtedly require an approach involving specifically targeting multiple, rather than single, signaling components in combination.

## Figures and Tables

**Figure 1 cells-08-01130-f001:**
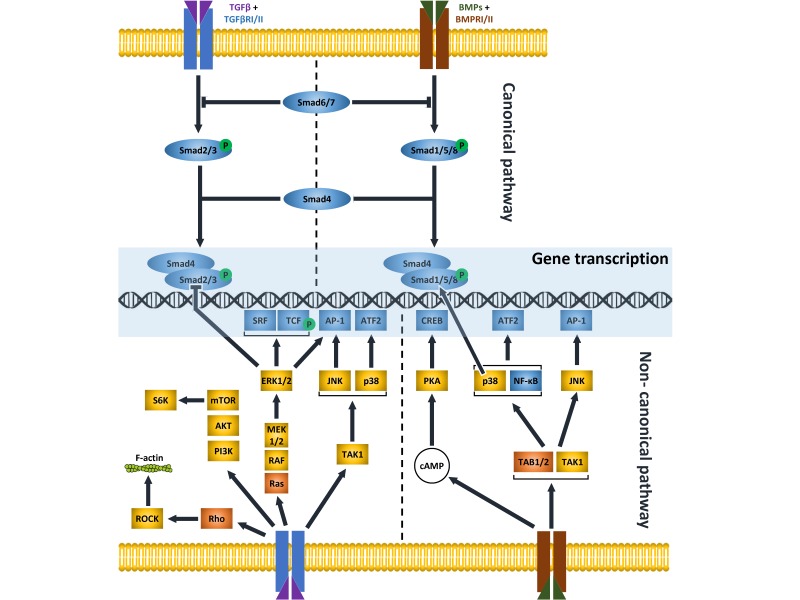
Essential overview of TGFβ and BMP signaling pathways. For simplicity, many details are omitted, and indirect connections are represented as single arrows. Transcription factors or DNA interacting proteins/kinases/other types of intracellular mediators, are represented as blue/orange/maroon boxes, respectively. SRF = Serum Response Factor; TCF = T Cell Factor; LEF = Lymphoid enhancer-binding factor; AP-1 = Activator Protein-1; ATF2 = Activating Transcription Factor 2; CREB = cAMP response element Binding Protein; NF-κB = nuclear factor kappa-light-chain-enhancer of activated B cells; ERK = Extracellular Signal Regulated Kinase; JNK = c-Jun N-terminal kinase; TAB1/2 = TGF-beta-activated kinase 1 and MAP3K7-binding protein 1/2; TAK1 = Mitogen-activated protein kinase kinase kinase 7.

**Figure 2 cells-08-01130-f002:**
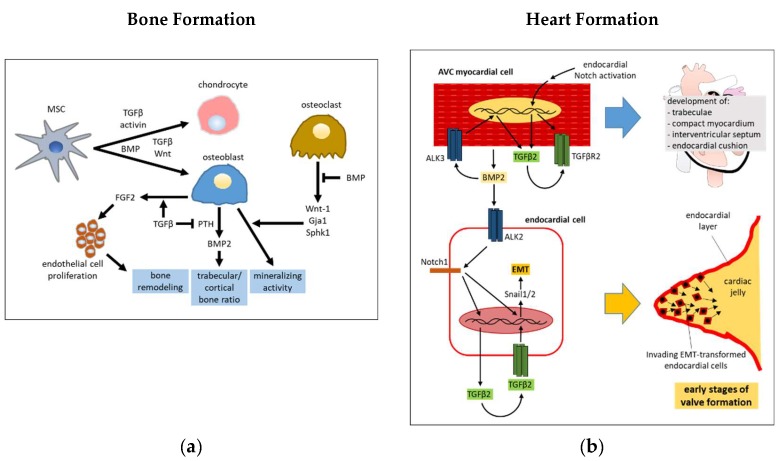
TGFβ and BMP interplay in bone and heart formation. (**a**) TGFβ can drive the differentiation of MSCs into chondrocytes or osteoblasts depending on the functional coupling with other factors, such as activin and Wnt, respectively. BMPs can induce the differentiation of these cells, but also block the production by osteoclasts of coupling factors that stimulate bone matrix synthesis activity by osteoblasts. TGFβR2 signaling is necessary to attenuate the PTH signaling that, in turn, enhances the production of BMP2, which can regulate the trabecular/cortical bone ratio. (**b**) AV myocardial cells require BMP2, TGFβ2, and activated Notch signaling status in endocardial endothelial cells to achieve multiple phases of heart development. BMP2 produced by myocardial cells induces the expression of Notch1 and TGFβ2 in endocardial endothelial cells. Notch1 and TGFβ2 signaling activate the EMT program that is required by these cells to invade the cardiac jelly in the early step of heart valve formation.

**Figure 3 cells-08-01130-f003:**
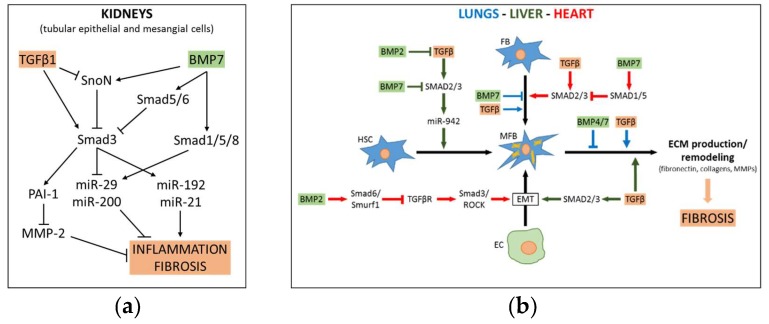
TGFβ and BMP interplay in kidney, lungs, liver and heart fibrosis. (**a**) In tubular epithelial cells TGFβ and BMP7 reciprocally oppose each other’s effects on fibrosis through regulating the expression of key miRNAs with antagonist functions. In mesangial cells BMP7 counteracts TGFβ pro-fibrotic activity via a SMAD5/6 mechanism that prevents SMAD3 nuclear accumulation, and subsequent MMP2 inhibition. (**b**) BMP2 can prevent the TGFβ-induced trans-differentiation of HSCs in MFBs through downregulating the expression of TGFβ itself and its receptors. By a more indirect mechanism (as in heart fibrosis), BMP2 can also counteract TGFβ-induced EMT. Arrows are colored according to organs of reference. HSC = hepatic stellate cell; FB = fibroblast; MFB = myofibroblast; EC = epithelial cell. Fibrosis limiting/promoting ligands are enclosed in green/orange boxes, respectively.

**Figure 4 cells-08-01130-f004:**
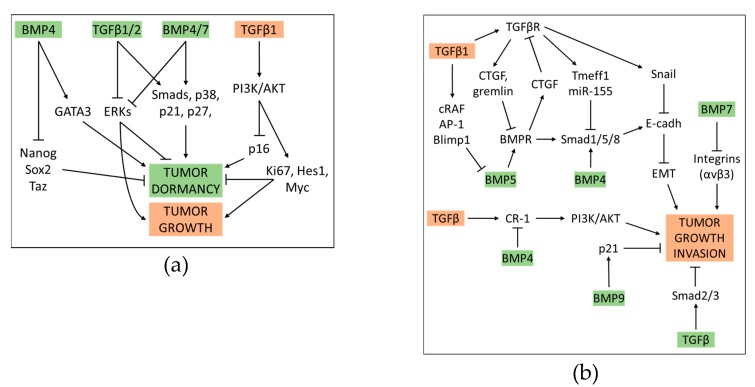
TGFβ and BMP interplay in tumor dormancy, growth and invasion. (**a**) TGFβ and BMPs pathways can cooperate to induce tumor dormancy. However, non-canonical TGFβ-activated PI3K/AKT signaling can promote the exit from dormancy and re-enable tumor cell proliferation. (**b**) TGFβ and BMPs play prevalently counteractive roles in tumor cell growth and invasion. The TGFβ/SMAD2 axis may induce cell cycle arrest in early phases of tumorigeneses (such as in HCC). However, at later tumor stages SMAD-independent TGFβ signaling is suggested to promote events that result in cell growth enhancement, the EMT, and invasiveness. Overall BMPs act to limit tumor cell proliferation and invasion, and TGFβ tumor-promoting actions may rely on the offsetting of BMPs dependent signals. Tumor limiting/promoting ligands are enclosed in green/orange boxes, respectively.

**Table 1 cells-08-01130-t001:** TGFβ and BMP ligands and related receptors.

Ligand	Type 1 Receptor	Type 2 Receptor	Coreceptor	Intracellular Mediator	Ref.
TGFβ1/2/3	ALK5	TβRII	endoglin	SMAD2/3	[9]
TGFβ1/2/3	ALK5	TβRII	betaglycan	SMAD2/3	[10]
TGFβ1/2/3	ALK1	TβRII	endoglin	SMAD1/5/8	[10]
TGFβ1/2/3	ALK1/2	ACVR2A, BMPR2		SMAD2/3, SMAD1/5/8	[10]
BMP1		BMPR1A			[11]
BMP2	ALK2/3	BMPR2, ACVR2A, ACVR2B		SMAD1/5/8, p38	[12,13]
BMP2	ALK3	BMPR2, ACVR2A, ACVR2B		SMAD2/3	[14]
BMP2	ALK2/3	BMPR2, ACVR2A		SMAD1/5/8	[15]
BMP4	ALK1/2	BMPR2		SMAD1/5	[16]
BMP4	ALK3	BMPR2		SMAD1/5, p38/ERK	[16]
BMP9/10	ALK1	BMPR2, ACVR2A, ACVR2B	endoglin	SMAD1/5/8	[17,18]
BMP6/7	BMPRI	ACVR2A, BMPR2		SMAD1/5/8, PI3K	[19]
BMP6/7	BMPRI	BMPRII		SMAD1/5/8	[20]
BMP6/7/9	ALK2	ACVR2A, ACVR2B		SMAD1/5	[21]
BMP2/6	Alk2	ACVR2A		SMAD1/5/8	[22]

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
