# Peer review of "The Interactivity between TGFβ and BMP Signaling in Organogenesis, Fibrosis, and Cancer"

_cells, 2019, doi:10.3390/cells8101130_

Round 1

Reviewer 1 Report

Dituri et al. submitted a short review article (5 pages long) entitled “The interactivity between TGFb and BMP signaling in organogenesis, fibrosis, and cancer”. This is such a broad and complex theme which will easily require an entire textbook length with multiple chapters and is almost impossible to summarize in one short review article. My biggest concern is that the authors may be oversimplifying this extremely complex biology. The authors should have picked up only one of the three themes (either organogenesis, fibrosis, or cancer) to discuss in detail in specific organs such as heart, lung, liver, and/or bone (or may pick up only one) in a consistent manner and to review numerous published original works of the interaction between TGF-b and BMP. There are a significant number of original papers published on this topic, some of which are not quoted in this manuscript.

Major Concerns:

Organogenesis is a developmental process that follows gastrulation consisting of cell division, cell differentiation, and overall integration to develop into functional organs. Each organ has unique developmental process that needs to be addressed in conjunction with TGF-b and BMP interaction. In the first section (1. TGFb and BMP…), what is lacking is a consistent discussion from the developmental biology viewpoint, including concepts of cell division, differentiation (e.g., EMT), morphogenesis, and their regulations. Authors should select the specific organs to discuss the role of TGF-b/BMP on organogenesis.

In the section of fibrosis (2. Role of TGFb-BMP…), there are multiple different pathological mechanisms to induce fibrosis depending upon the etiology and organs. For example, likely etiology of fibrosis in heart and kidney may be quite different; myocardial fibrosis frequently follows ischemia or hemodynamic overload, whereas renal fibrosis mainly follows abnormal immunological process or inflammation. Organ-specificity (or difference among organs) was not discussed at al. Addition of a table to compare the fibrotic process in different organs may be helpful, incorporating etiology, mechanisms, and the involvement of TGF-b/BMP in different organs with citation of original papers. With regard to myocardial fibrosis, there are complex interactions between fibroblasts, endothelial cells, cardiomyocytes, and inflammatory cells. TGF-b is certainly a primary growth factor that promotes fibrosis, which may be predominantly produced by active fibroblasts, but the effector cells are variable. These complexities are not addressed in the manuscript.

The involvement of TGF-b in cancer development is context dependent, including background organs, originating cell types and pathological classification, and stages of cancer. Regulation of TGF-b in cancer development is multifold including cell types, extracellular environment (stroma), angiogenesis, and BMP and other signaling pathways. These complexity is not clearly introduced in the manuscript.

Minor Comments

There should be consistent format to address TGF-b. In the text, both TGF-b and TGFb are used randomly.

Figure 1 is not legible. It is hard to recognize what are shown in the figure.

The authors should more effectively paragraphing the text. For example, in the second section describing fibrosis, the text should be subdivided into kidney (line 101 to 131), liver (line 131 to line 146), lung (line 146 to 155), heart (line 155 to 167), and summary (the rest). The same principle applies to third section.

Several important statements in the text are not accompanied by appropriate citations. For example, there is no single citation listed from line 26 to 35. In a review article, authors are encouraged to quote original research papers than review articles.

Author Response

Dituri et al. submitted a short review article (5 pages long) entitled “The interactivity between TGFb and BMP signaling in organogenesis, fibrosis, and cancer”. This is such a broad and complex theme which will easily require an entire textbook length with multiple chapters and is almost impossible to summarize in one short review article. My biggest concern is that the authors may be oversimplifying this extremely complex biology. The authors should have picked up only one of the three themes (either organogenesis, fibrosis, or cancer) to discuss in detail in specific organs such as heart, lung, liver, and/or bone (or may pick up only one) in a consistent manner and to review numerous published original works of the interaction between TGF-b and BMP. There are a significant number of original papers published on this topic, some of which are not quoted in this manuscript.

Answer. We also believe the subjects of this review are potentially requiring a very long treatise. Of course we could not ever have covered the entire relative knowledge, but, rather we tried to focus on some significant circumstances where the interplay between TGFb and BMPs offers an example of how ligands belonging to the same superfamily can interact in cooperative or counteractive manner, depending on the organ/disease, to influence physiological or pathologic processes.

Major Concerns:

Organogenesis is a developmental process that follows gastrulation consisting of cell division, cell differentiation, and overall integration to develop into functional organs. Each organ has unique developmental process that needs to be addressed in conjunction with TGF-b and BMP interaction. In the first section (1. TGFb and BMP…), what is lacking is a consistent discussion from the developmental biology viewpoint, including concepts of cell division, differentiation (e.g., EMT), morphogenesis, and their regulations. Authors should select the specific organs to discuss the role of TGF-b/BMP on organogenesis.

Answer. We have divided the section of organogenesis into subsection, emphasizing the role of TGFb/BMPs in two paradigmatic organs/tissues, namely bone/cartilage and heart valve formation. An explanatory figure (2) has been introduced also.

In the section of fibrosis (2. Role of TGFb-BMP…), there are multiple different pathological mechanisms to induce fibrosis depending upon the etiology and organs. For example, likely etiology of fibrosis in heart and kidney may be quite different; myocardial fibrosis frequently follows ischemia or hemodynamic overload, whereas renal fibrosis mainly follows abnormal immunological process or inflammation. Organ-specificity (or difference among organs) was not discussed at al. Addition of a table to compare the fibrotic process in different organs may be helpful, incorporating etiology, mechanisms, and the involvement of TGF-b/BMP in different organs with citation of original papers. With regard to myocardial fibrosis, there are complex interactions between fibroblasts, endothelial cells, cardiomyocytes, and inflammatory cells. TGF-bis certainly a primary growth factor that promotes fibrosis, which may be predominantly produced by active fibroblasts, but the effector cells are variable. These complexities are not addressed in the manuscript.

Answer. The section related to fibrosis has been divided into five parts (1- introductive; 2- liver; 3- renal; 4-pulmonary; 5- myocardial). A figure has been added to illustrate some main molecular aspects involving TGFb/BMPs in the regulation of fibrosis. A discussion of all the aspects rightly requested by the reviewer would have enormously increased the quantity of useful information, but also unsustainably the length of the manuscript. The role of epithelial to mesenchymal transition as well as of activated fibroblasts and other cell types have also been discussed.

The involvement of TGF-b in cancer development is context dependent, including background organs, originating cell types and pathological classification, and stages of cancer. Regulation of TGF-b in cancer development is multifold including cell types, extracellular environment (stroma), angiogenesis, and BMP and other signaling pathways. These complexity is not clearly introduced in the manuscript.

Answer. We have added some information regarding the cancer stage-related tumor supporting/promoting activities of TGFb relatively to HCC, and tried to discussed the context-dependence in relation to interaction TGFb/BMPs. The information the reviewer would have liked to find on the multiple aspects of TGFb biology in cancer are undoubtedly relevant to enrich and strengthen the knowledge, and have been widely described in many excellent reviews so far.  Unfortunately, a so in depth description would have enormously increased the length of the manuscript. A figure has been added on molecular signaling affected by TGFb/BMPs in cancer.

Minor Comments

There should be consistent format to address TGF-b. In the text, both TGF-b and TGFb are used randomly.

Answer. We have addressed this point.

Figure 1 is not legible. It is hard to recognize what are shown in the figure.

Answer. This figure depicts the signaling of TGFb/BMPs, showing some of major molecular effectors/switches belonging to these pathways. Transcription-DNA-interacting factors, kinases, other intracellular mediators, are represented with different colors (as now explained in the figure legend)

The authors should more effectively paragraphing the text. For example, in the second section describing fibrosis, the text should be subdivided into kidney (line 101 to 131), liver (line 131 to line 146), lung (line 146 to 155), heart (line 155 to 167), and summary (the rest). The same principle applies to third section.

Answer. The paragraphing of fibrosis section has been done, as explained before. However, we regret having found hard to paragraph the cancer section, since a number of described combined actions of TGFb/BMPs belong to molecular mechanisms are shared by diverse cancer types.

Several important statements in the text are not accompanied by appropriate citations. For example, there is no single citation listed from line 26 to 35. In a review article, authors are encouraged to quote original research papers than review articles.

Answer. The number of references is now almost doubled. The references the reviewer refers to were grouped together at the end of the relative text.

Reviewer 2 Report

TGF-β and BMP pathways are highly studied pathways regarding every function they take part in, from embryogenesis to cancer and death. There are brilliant highly cited reviews on each topic so the task of writing another review is challenging. The authors have tried to write one but there are certain things that are evidently lacking in the review.

1.       Critical reviews on the topic that should have been quoted are missing significantly and that is evident from the very first paragraph, for e.g. PMID: 29643418, PMID: 22298955, PMID: 30268436

2.       The authors, under the heading organogenesis, have focused mainly on bone formation and its defects, I wonder how the authors missed the review PMID: 22298955 which has an extensive literature on bone formation including a number of TGF beta conditional knockout mouse models.

3.       The role of TGF beta/BMP pathways in stem cells and their differentiation, very important topics in organogenesis, regeneration and metastasis, are also missing. Refer - PMID: 28108485, PMID: 22710171, PMID: 24487640.

4.       I think the role of TGF beta/BMP pathways in modulating the microenvironment that plays significant role in fibrosis and cancer is also missing.

5.       Important reviews regarding Cancer are also missing e.g. - PMID: 29701666; 18662538; 28918914, 27039259

Author Response

TGF-β and BMP pathways are highly studied pathways regarding every function they take part in, from embryogenesis to cancer and death. There are brilliant highly cited reviews on each topic so the task of writing another review is challenging. The authors have tried to write one but there are certain things that are evidently lacking in the review.

Critical reviews on the topic that should have been quoted are missing significantly and that is evident from the very first paragraph, for e.g. PMID: 29643418, PMID: 22298955, PMID: 30268436

Answer.  The review PMID: 29643418 (ref. #8) has been introduced in the text; The review PMID: 22298955 (ref. #35) has been cited in the text, and 2 references cited by this review have also been included (refs. #43 and 44); The study PMID: 30268436 has been cited (ref. #122).

The authors, under the heading organogenesis, have focused mainly on bone formation and its defects, I wonder how the authors missed the review PMID: 22298955 which has an extensive literature on bone formation including a number of TGF beta conditional knockout mouse models.

Answer.  The review PMID: 22298955 (ref. #35) has been cited and discussed in the text (as in the previous answer). Moreover, two references cited within this review have been discussed (please see #43 and #44).

The role of TGF beta/BMP pathways in stem cells and their differentiation, very important topics in organogenesis, regeneration and metastasis, are also missing. Refer - PMID: 28108485, PMID: 22710171, PMID: 24487640.

Answer.  The reviews have been cited (refs. #23 and #24 and #39, respectively).

I think the role of TGF beta/BMP pathways in modulating the microenvironment that plays significant role in fibrosis and cancer is also missing.

Answer.  The text relative to role of TGF beta/BMP in stromal microenvironment in fibrosis and cancer has been increased by the addition of information about hepatic stellate cells, myofibroblasts, and EMT.

Important reviews regarding Cancer are also missing e.g. - PMID: 29701666; 18662538; 28918914, 27039259

Answer.  The following PMID: 29701666; 18662538; 28918914 have been cited (refs. #115, #114, #121, respectively).

Reviewer 3 Report

In this manuscript, Dituri et al have reviewed the literature regarding the crosstalk of TGFB and BMP signaling in multiple biological processes. The chosen area of review is very broad, therefore, its almost impossible to comprehensibly review the role of these pathways in the regulation of fibrosis, cancer and organ development. Also, the present manuscript is very descriptive, its need to include illustrations and tables to present the concepts to readers in a more organized way. I have the following comments.

1.       Authors have chosen such a broad topic to review, therefore, multiple areas have been touched superficially. This reviewer is afraid to state that fixing this issue will need a major modification.

2.       An introduction section needs to be included. This section should describe the key players of the TGF and BMP pathways. Preferably, authors should present a systematic table listing the ligands, receptors, intracellular players and downstream transcription factors in a separate column/rows. The introduction should focus on the basic operation of the pathways. Following sections will further expand to discuss the application in various biological processes (Development, fibrosis, and cancer)

3.       All three section (section 1 to 3 in current form), specifically, Development, fibrosis, cancer need to present table listing all the major original studies of these pathways describing crosstalk. To address this, authors need to include a table in each section (total 3 tables). These three tables will be key for the readers to quickly find the major original reports.

4.       In continuation of the above comment, all three sections need to be supported by illustrations.

In summary, this reviewer truly believes that the addition of above-mentioned tables and illustrations will significantly strengthen the manuscript.

Author Response

In this manuscript, Dituri et al have reviewed the literature regarding the crosstalk of TGFB and BMP signaling in multiple biological processes. The chosen area of review is very broad, therefore, its almost impossible to comprehensibly review the role of these pathways in the regulation of fibrosis, cancer and organ development. Also, the present manuscript is very descriptive, its need to include illustrations and tables to present the concepts to readers in a more organized way. I have the following comments.

Authors have chosen such a broad topic to review, therefore, multiple areas have been touched superficially. This reviewer is afraid to state that fixing this issue will need a major modification.

Answer: We acknowledge the rightness of this observation. Because of this, we have tried to extend the overall discussion, and have also added more references to support the discussion.

An introduction section needs to be included. This section should describe the key players of the TGF and BMP pathways. Preferably, authors should present a systematic table listing the ligands, receptors, intracellular players and downstream transcription factors in a separate column/rows. The introduction should focus on the basic operation of the pathways. Following sections will further expand to discuss the application in various biological processes (Development, fibrosis, and cancer).

Answer.  An introduction on TGFb and BMP pathway is now present. A table with ligands, receptors and intracellular mediators for TGFb/BMPs (with references included) has been added.

All three section (section 1 to 3 in current form), specifically, Development, fibrosis, cancer need to present table listing all the major original studies of these pathways describing crosstalk. To address this, authors need to include a table in each section (total 3 tables). These three tables will be key for the readers to quickly find the major original reports.

Answer.  We regret having found hard to satisfy this point due to the difficulty to build tables that effectively can be useful for the reader, especially because of the need to more extensively describe the related studies, and at the risk of extending the explanatory text beyond the limit commonly acceptable for a table.

In continuation of the above comment, all three sections need to be supported by illustrations.

Answer.  Three figures have been added, one related to each section.

Round 2

Reviewer 1 Report

The authors submitted the first revision of the above review article. My initial critiques include a) very broad theme with a short text length, b) some deficiency of necessary references, and c) somewhat superficial view of very complex biology. The revised text reads better with an addition of texts, figures, and critical references. The format of the text has been improved, as well. However, there are still some issues that need to be taken care of. Especially, figures require further improvement in terms of accuracy and clarity. All abbreviations need to be spelled out at the first encounter. The followings are my specific comments. I suggest the entire text should be reviewed by the professional editorial service before resubmission.

L 40: These abbreviations need to be spelled out.

Figure 1: It is still hard to read the letters in the figure. Letters can be better shown in a black color. Abbreviations need to be expressed.

L 56: FGF, needs to be spelled out.

L 99 to L 111: The text and the Figure 2 (left) should be concordant. Or figure legends should be added to explain what is presented in the figure.

Figure 2: Full spelling of MSC should be included. A: Bone formation, B: Heart valve formation. In bone development, the figure should summarize what is discussed in the text.

L 117-133: There are both endothelial cells and endocardial cells in this section. In the figure, endocardial cell is used. Please use the word consistently if you are referring to the same concept (endocardial cells or endocardial endothelial cells).

L 146-148: Needs references.

L 175: ad -> and

L 204: TAA needs to be spelled out.

L 248: THR-123 needs to be spelled out.

Figure 3: A. KIDNEYS: This diagram is confusing as it is not clear where these interactions take place. Do these occur in fibroblasts or else? Multiple different cells are involved in this process and all participant cells should be introduced. B. LUNGS-LIVER-HEART: This is also a confusing diagram. Is it true that there is a direct molecular interaction (inhibitory effect) between BMP2 and TGFb? During EMT, is TGFb not involved in direct canonical pathway (R-Smad pathway including Smad2/3 activation), as is shown in this diagram? Both diagrams require revision for better accuracy.

L 357 to L 415: This paragraph is too long. I suggest that this long paragraph divided into a few paragraphs.

L 362: K167, HES1, and MYC need to be spelled out.

Figure 4: It is not clear what these two diagrams stand for. These two diagrams require detailed figure legends.

Author Response

Dear reviewer,

we have make an effort to satisfy you requests. 

1) The text has been extensively reviewed to correct grammar and syntax errors. Small text parts have been added to 

2) The abbreviations have been spelled out as requested.

3) The figure 1 is now appearing as a new version with letters in black color.

4) Both the panels of Figure 2 have been revisited and enriched to better summarize what described in the text. The relative legend has also been extended.

5) The Figure 3 has been revisited and enriched for more clarity. The legend of the figure has also been extended.

6) A more detailed legend has been added to Figure 4 to help the interpretation of diagrams shown.

Hope this will suffice.

Kind Regards 

Francesco Dituri.

Reviewer 2 Report

Authors have significantly improved the manuscript especially the presentation of the conceptual aspects.

A little detail about the micro-environment and knockout models still seem missing.

Overall, the manuscript reads much better.

Author Response

Dear Reviewer, 

We have extensively reviewed the manuscript and some figures to ameliorate the format and the clarity of discussed concepts. Short text parts have also been added.

In the Figure 3b the role of microenvironmental cells (fibroblasts, hepatic stellate cells and myofibrblasts) in fibrosis has been highlighted with more detail. 

Actually we could not cite some further papers on conditional knockouts, as you suggested in the previous revision (PMID: 22298955) because the part in this review that specifically refers to the interaction between TGFb and BMP in bone development seems very short (5 text lines) and we think does not add significant information for the purpose of this review. Anyway this review (ref. 35) and one internal reference (ref. 43) have been mentioned.

Kind Regards 

Francesco Dituri.